# The Role of Long Non-Coding RNAs (lncRNAs) in Female Oriented Cancers

**DOI:** 10.3390/cancers13236102

**Published:** 2021-12-03

**Authors:** Faiza Naz, Imran Tariq, Sajid Ali, Ahmed Somaida, Eduard Preis, Udo Bakowsky

**Affiliations:** 1Punjab University College of Pharmacy, Allama Iqbal Campus, University of the Punjab, Lahore 54000, Pakistan; faizanazrph@gmail.com; 2Department of Pharmaceutics and Biopharmaceutics, University of Marburg, Robert-Koch-Str. 4, 35037 Marburg, Germany or sajid.ali@kemi.uu.se (S.A.); somaida@staff.uni-marburg.de (A.S.); eduard.preis@pharmazie.uni-marburg.de (E.P.); 3Angström Laboratory, Department of Chemistry, Uppsala University, 75123 Uppsala, Sweden

**Keywords:** lncRNAs, breast cancer, gynecological cancers, HOTAIR, NEAT1, H19, MALAT1, MEG3

## Abstract

**Simple Summary:**

Breast and gynecological cancers, broadly termed as female-oriented cancers, are the primary cause of death among females in developed and developing countries. Tumor invasion and metastasis cause the aggressiveness of these cancer types. The occurrence and frequency of women’s cancers are associated with genetics, personal lifestyle, body shape, age, menopause status, history of exposure to carcinogens or viruses, and geographical habitat. Moreover, ncRNAs, especially lncRNAs, play an essential role in regulating cellular functions within such cancers. LncRNAs can play dual roles. They can either exert tumor-suppressive or oncogenic functions in women’s cancers. Accumulating evidence suggests that lncRNAs can be promising prognostic and diagnostic biomarkers and therapeutic targets in cancers. Thus, understanding the mechanism and role of lncRNAs might provide new opportunities for diagnosing and treating female-oriented cancers. In this review, we discuss the worldwide incidence of breast and gynecological cancers, including endometrial, cervical, ovarian, vaginal, vulvar cancers, and GTN among women. We further provide various perspectives on the association of some lncRNAs, i.e., HOTAIR, NEAT1, H19, MALAT1, and MEG3, in terms of invasion, proliferation, metastasis, apoptosis, and drug resistance of breast and gynecological cancers based on recent discoveries. Finally, we present insight and prospects into the potential of these lncRNAs for evaluating the prognosis, diagnosis, and treatment of such cancers.

**Abstract:**

Recent advances in molecular biology have discovered the mysterious role of long non-coding RNAs (lncRNAs) as potential biomarkers for cancer diagnosis and targets for advanced cancer therapy. Studies have shown that lncRNAs take part in the incidence and development of cancers in humans. However, previously they were considered as mere RNA noise or transcription byproducts lacking any biological function. In this article, we present a summary of the progress on ascertaining the biological functions of five lncRNAs (HOTAIR, NEAT1, H19, MALAT1, and MEG3) in female-oriented cancers, including breast and gynecological cancers, with the perspective of carcinogenesis, cancer proliferation, and metastasis. We provide the current state of knowledge from the past five years of the literature to discuss the clinical importance of such lncRNAs as therapeutic targets or early diagnostic biomarkers. We reviewed the consequences, either oncogenic or tumor-suppressing features, of their aberrant expression in female-oriented cancers. We tried to explain the established mechanism by which they regulate cancer proliferation and metastasis by competing with miRNAs and other mechanisms involved via regulating genes and signaling pathways. In addition, we revealed the association between stated lncRNAs and chemo-resistance or radio-resistance and their potential clinical applications and future perspectives.

## 1. Introduction

The long non-coding RNAs (lncRNAs) are one of two basic classes of the non-coding RNAs (ncRNA), arbitrarily defined as ncRNA transcripts with at least 200 base pairs (bp) that do not encode proteins [1]. It has been estimated from ENCODE project data that around 70–80% of the human genome undergoes transcription, but out of it, only 2% genome codes for proteins. The remaining transcribed RNAs are ncRNAs whose functions have not been completely recognized yet. However, it is seen to be involved in the regulation of gene expression at transcriptional, post-transcriptional, or translational levels and various other diversified functions in cells as well [2,3].

By next-generation sequencing, tens of thousands of lncRNA loci have been identified from eukaryotes to humans. The current Gencode annotation approximates nearly 17,952 lncRNA genes and 48,438 transcripts in humans, widely associated with cellular processes during normal development and physiology [4,5]. They are very heterogeneous as their transcripts may carry nucleotides from several hundred to several thousand [6]. Like mRNAs, they are 5′-capped, polyadenylated, often spliced, and usually transcribed by RNA polymerase II and III [7]. The lncRNAs are found within the cytoplasmic or nuclear fractions [8,9]. Cytoplasmic lncRNAs are generally involved in regulating mRNA turnover, alternative splicing, post-transcriptional regulation, protein stability, microRNA sponging, and the regulation of signaling pathways [10]. Nuclear lncRNAs, however, regulate the cellular processes by interacting with chromatin architecture [11].

Usually, lncRNAs do not have codon preservation constraints and demonstrate only modest primary sequence conservation. Thus, such sequences of lncRNAs seem to be less significant than their secondary or tertiary structures, and in most cases, their functions are seen to be reliant on their structural conservation rather than sequence conservation [12,13]. Previously, lncRNAs were thought to lack the protein-coding ability, but currently, it is found that they can have short open reading frames (sORFs) [14]. Recent studies have discovered a variety of lncRNAs with sORFs that can encode small functional peptides in the human body, which can regulate cellular metabolism, muscle function, and the suppression of cancer growth [15].

Despite the normal physiological regulation, disruption of lncRNA’s expression has also been intrinsically linked with the occurrence and development of a range of diseases, including cancer. Hence, it led to the raised interest in studying lncRNAs with the prospect of discovering novel therapeutic and diagnostic strategies [16]. Worldwide, both in high-income or middle-income countries, cancer represents one of the most common causes of death among women. Risk factors for female-oriented cancers can be classified into two main classes: cancers associated with exposures also present in men, and forms of cancers unique to women [17]. Moreover, changes in reproductive patterns, such as later age at first childbirth and lower parity, have contributed to higher female cancer burden worldwide [18]. Several female organs are prone to be affected by different types of cancers. Cancers of the reproductive system, such as the ovarian, uterine, cervical, fallopian tube, vaginal, vulvar, and gestational trophoblastic cancers, can be summarized as gynecological cancers [19,20]. Collectively gynecologic and breast cancers can be termed as female-oriented cancers.

The expression profiles of lncRNAs are considered important in female-oriented cancers. Many of the lncRNAs are revealed to be involved in various imperative processes of such cancers, including genesis, cancer development, proliferation, invasion, metastasis, and drug resistance, while some others exhibited inhibition of these processes. Thus, by demonstrating either tumorigenesis or tumor-suppressive roles, the aberrant expression of lncRNAs can significantly contribute to the development of such cancers [21]. At present, 172,216 lncRNA transcripts and 96,308 lncRNA genes, influencing a wide variety of cellular biological processes and the status of cancers in females, have been identified [22].

Previously, the involvement of various lncRNAs has been studied individually by multiple researchers in breast cancer and various gynecological cancers in women, but no one discussed it mutually in the perspective of female-oriented cancers. Therefore, in this article, we highlight the role of various lncRNAs and specifically five lncRNAs—HOTAIR, NEAT1, H19, MALAT1, and MEG3—in cancers unique to women, including breast and gynecological cancers.

## 2. Incidence of Female-Oriented Cancers

Worldwide, cancer is considered one of the most common causes of death in females. According to WHO statistics, the incidence of breast cancer is much higher than any other women’s cancer. However, they are spreading with increased rates day by day (Figure 1). The determination and early detection of cancer and the discovery of new and more effective treatment strategies might help control morbidity and mortality rates.

### 2.1. Breast Cancer (BC)

Breast cancer (BC) is the most diagnosed form, with over 2 million cases reported annually, whereas lung cancer, in comparison, is the second most common cause of death because of the malignancy of cancerous cells [23]. Ductal hyperproliferation usually causes the onset of breast tumors, which leads to benign tumors or even metastatic carcinomas because of the interplay of persistent stimulation and different carcinogenic factors. As a part of tumor microenvironments, stromal cells and macrophages play a vital role in breast cancer development and progression [24]. Macrophages can promote angiogenesis and empower cancerous cells’ immune resistance by creating a mutagenic inflammatory microenvironment [25]. Similarly, epigenetic modifications such as DNA methylation in the tumor microenvironment have been detected to enhance the risk of carcinogenesis [26]. Moreover, cancer stem cells (CSCs), having self-renewal abilities, are observed to be associated with BC initiation, immune escape, recurrence, and exhibited resistance to chemo and radiotherapy [27].

### 2.2. Gynecological Cancers

#### 2.2.1. Endometrial Cancer (EC)

Different organs of the female reproductive system are also affected by cancer. The most prevalent gynecological cancer is endometrial cancer (EC) of the uterus, with over 61,000 women diagnosed each year in the USA [28], and an estimated 320,000 new cases and 76,000 deaths reported worldwide in 2012 [29]. EC cultivates in the uterine inner lining from the glandular epithelial sheet, which covers the luminal surface and releases substances needed for normal menstruation or embryonic development [30]. In addition to obesity, altered hormonal levels, reproductive factors, and genetic predispositions are also recognized as EC’s major risk factors. EC is genetically heterogeneous, apart from a patient’s subgroup belonging to a cancer predisposition syndrome, i.e., Lynch Syndrome, triggered by the germline alterations of DNA mismatch repair genes such as MLH1, PMS2, MSH2, and MSH6. EC also exhibits a high frequency of several other germline mutations in cancer predisposition genes [31].

#### 2.2.2. Cervical Cancer (CC)

Globally, CC is statistically calculated as the fourth most prevalent form of cancer plus the fourth foremost cause of cancer deaths in females, right after breast, colorectal, and lung cancers. According to the data of Lancet Global Health, nearly 570,000 females faced cervical cancer, of which 311,000 females died in 2018 [32]. Infections with high-risk human papillomaviruses (HPV) are the leading cause of CC. Thus, HPV screening and anti-HPV vaccination programs are considered effective disease prevention strategies [33]. The histological subtypes of CC are adenocarcinoma (25%) and squamous cell carcinoma (70%) [34]. While recognizing the pre-cancerous lesions, a Pap smear test can help identify early changes in cervical epithelium and the early stage of invasive CC [35].

#### 2.2.3. Ovarian Cancer (OC)

OC is relatively rare compared to other women’s cancers but poses the highest cancer mortality rate. The incidence of OC has been estimated at 11.7–12.1 per 100,000 in Europe and USA, with a lower disease rate in the Middle East and Asia [36]. OC’s pathology encompasses a heterogeneous group of malignancies originating in the germ cells, epithelial cells, fallopian tube, and mesenchyme that differ in etiology, molecular biology, and several other characteristics. However, approximately ninety percent of OCs have epithelial origin [37]. Most patients (60%) are diagnosed with advanced disease progression, leading to a significant mortality rate. Thus, improved prevention and early detection have always been the research priority, as early disease diagnosis results in a higher survival rate (93%) [38].

#### 2.2.4. Primary Fallopian Tube Carcinoma (PFTC)

PFTC is rare and causes around 0.14–1.8% of all gynecological malignancies. However, PFTC is similar to OC in clinical diagnosis and cannot be distinguished pre-operatively [39]. PFTC seems like a cystic-solid or a solid adnexal mass that looks like an epithelial ovarian cancer (EOC). Based on preoperative images, it may lead to increased chances of being misdiagnosed as EOC. However, MRI has been advanced to differentiate PFTC from EOC by recognizing the distinctive features of PFTC, such as hydrosalpinx, hydrosalpinx with mural papillary nodules, and intrauterine fluid accumulation [40].

#### 2.2.5. Vaginal Cancer (VC)

VC is relatively uncommon, comprising about 3% of all gynecologic malignancies, and nearly 3000 cases are diagnosed in the United States annually, with approximately 900 deaths [41]. Mainly, VC occurs in older or postmenopausal women because of high-risk HPV infection; however, it has also been reported in younger women [42].

#### 2.2.6. Vulvar Cancer (VC)

VC is considered the twentieth most prevalent cancer in women, with estimated 6190 cases reported in 2018, and the number is still increasing at the rate of 0.6% annually, but with a comparably decreased survival rate. This disease is mainly diagnosed in older women with a median age of 68 years. Vulvar squamous cell carcinoma (VSCC) is seen to be the utmost common malignancy among female vulvar cancers, constituting above 90% of all vulvar malignancies of females [43,44]. It is usually believed that VSCC has two etiological pathways, i.e., genetic alterations including p53 mutations or p16INK4a silencing and high-risk HPV-dependent routes [45].

#### 2.2.7. Gestational Trophoblastic Neoplasia (GTN)

The development of malignant cancerous cells after any type of pregnancy is known as GTN, a rare gynecological cancer with 2 cases in 1000 pregnancies reported in Japan and Southeast Asian countries, while 1 in 1500 pregnancies in the United States [46]. GTN occurs during pregnancy and is characterized as benign lesions produced by abnormal trophoblastic proliferation. GTN consists of a wide array of interdependent tumors which develop after abnormal reproduction of trophoblastic tissues. Some examples are invasive hydatidiform mole (HM), epithelioid trophoblastic tumor (ETT), malignant choriocarcinoma tumors as well as placental site trophoblastic tumors (PSTT) [47,48].

## 3. LncRNAs: Classification and Biogenesis

### 3.1. Classification

lncRNAs are often classified into two major categories, (A) linear lncRNAs and (B) circular lncRNAs (circRNAs), as mentioned in Figure 2 [49].


*1*.
*Linear lncRNAs*
Linear lncRNAs are classified based on their genomic localization and interaction with protein-coding genes, presence of accompanying repeat elements, the likeness with mRNA as well as function. Based on their genomic location and their direction of transcription to complementary protein-coding genes, lncRNAs are classified into the following categories [50]:(*A*)*Intronic lncRNA*(i)Sense intronic lncRNAs are located inside the intron of the protein-coding gene and transcribed from a coding strand of the respective gene. For example, lncGHRL3:3 is believed to be an intronic and sense overlapping lncRNA. It is located at chromosome 3 within the ghrelin gene, probably involved in the regulation of T2DM, and can be presumed as a potential biomarker of T2DM [14].(ii)Antisense intronic lncRNAs also exist in the intron region of a protein-coding gene but are transcribed from the opposite direction of the coding strand of that gene. For example, ANRASSF1 is an intronic antisense lncRNA that inhibits the function of the tumor suppressor gene RASSF1A by targeting its promoter region [51].(*B*)*Intergenic lncRNA*(i)Sense intergenic lncRNAs are sited between two protein-coding genes with overlapping sense strands of the coding gene. LincRNA-p21 is an important example induced during DNA damage by p53 (a tumor suppressor protein) to relay its anti-oncogenic functions [52].(ii)Antisense intergenic lncRNAs are located between two coding genes while transcribed from the antisense strand of a protein-coding gene, e.g., HOTAIR lncRNA. It belongs to a subclass of lincRNA that can decrease radiosensitivity in laryngeal cancer patients by regulating miR-454-3p [53].(*C*)*Exonic lncRNA*(i)Sense exonic lncRNAs are synthesized by transcribing lncRNA sequence from the sense strand of protein-coding gene and comprise the exons of that gene. For example, NONHSAG044354 is a sense exonic lncRNA associated with inflammatory bowel diseases gene BACH2 [54].(ii)Antisense exonic lncRNA is produced if its transcript is derived from the intron of the protein-coding gene on the opposite strand, e.g., Tsix lncRNA. Tsix lncRNA is an anti-sense lncRNA to the Xist gene, known to orchestrate the X-inactivation during dosage compensation by inhibiting the expression of Xist RNA and thus influences the random choice of which X will be inactivated [55].
(*D*)
*Enhancer RNAs (eRNAs)*




*Enhancer RNAs (eRNAs)* originate from enhancer regions of coding genes and facilitate the positioning of transcription factors into the promoters of protein-coding genes [56]. PVT1 lncRNA is an eRNA transcribed from the PVT1 locus and has been proved to have an oncogenic function by regulating the MYC gene expression [57]. However, emerging studies pointed out that eRNAs are distinct from lncRNAs, although these species may share a similar role in activating mRNA transcription. They showed that unlike the promoters of lncRNAs, enhancers do not show any bias in the direction of transcription initiation. Moreover, lncRNAs go through splicing and polyadenylation maturation processes, while shorter eRNAs (2 kb) show little evidence of consistent splicing and polyadenylaion. eRNAs may facilitate activation of promoter-driven transcription or enhancer-promoter interaction [58,59].

Based on the way they perform their regulatory function, lncRNAs are classified as: *(A) Cis-acting lncRNAs* influencing the chromatin state or the expression of nearby genes, such as Kcnq1ot1 and the Xist lncRNAs which regulate gene expression in cis [60] or *(B) Trans-acting lncRNAs* leaving the transcription site and executing cellular functions in trans, i.e., on distant genes [61]. HOTAIR is one of the first reported lncRNAs which regulate gene expression in trans. It is expressed from the HOXC locus and repress the transcription from distant HOXD cluster by recruiting PRC2 [62]. Moreover, whether in a cis or trans way, lncRNAs bring out their gene regulatory activities either as signal lncRNAs, guide lncRNAs, decoy lncRNAs, or scaffold lncRNAs [63].


*2*.
*Circular lncRNAs (circRNAs)*



CircRNAs are recognized as lncRNAs that form a circle via back splicing of one or more exons named as extra-coding RNAs (ecRNAs), or one or more introns termed as circular intronic RNAs (ciRNAs), or both intron and exon fragments of the parent gene. Such circRNAs are more stable than linear ncRNAs since their circular structure is resistant to degradation by RNA decay machinery [64].

### 3.2. Biogenesis

Some lncRNAs or even classes of lncRNAs are regulated differentially during their biogenesis (as shown in Figure 2), as the biogenesis of lncRNAs is thought to be stage-specific and cell type-specific and controlled by stage and cell type-specific stimuli [65]. Different classes of lncRNAs are transcribed from either the exonic or intergenic area or the distant protein-coding areas of the genome, usually by RNA-polymerase II enzyme or RNA-polymerase III. Then, the pre-mature lncRNAs get capped at 5′-end with methyl-guanosine and 3′- end polyadenylated [66]. Further, they undergo alternative splicing and RNA editing processes to generate diversity [67]. As a final point, mature lncRNAs are released and transported to other cellular sites based on their requirements in various cellular activities [68]. Epigenetic modification is involved in the biogenesis of lncRNAs; for instance, H3K4 methylation induces the transcriptional activation of the genes, while H3K27 tri-methylation directs gene silencing [69]. For the deep identification of functions and biogenesis of lncRNAs, various techniques including CLIP, RAP, CHART, ChIRP-Seq, RIP, CRISPR, and RNA pull-down are being used [70].

## 4. LncRNAs and Regulatory Implications

Characterization and analysis of functional pathways that the lncRNAs are involved with demonstrate that lncRNAs interact with the chromatin, with the RNA, or the protein to exhibit their effects to modulate migration, proliferation, differentiation, cell death, and apoptosis [71]. In general, they regulate gene expression in cancerous cells by either altering chromatin structure, activating or silencing a gene or a gene family, and in some cases, whole chromosomes via cis- or trans-methods [72]. Moreover, an essential regulatory aspect of lncRNAs is their association with the epigenetic machinery and the recruitment of its regulatory apparatus to specific loci, leading to the DNA methylation and/or post-translational modifications of histones. Aberrant expression of such lncRNAs, which interact with epigenetic modifiers, leads to severe epigenetic disruption and thus altered gene expression, cellular dysregulation, and malignant transformation. For example, HOTAIR causes the BC cell invasion and metastasis by interacting with and causing genomic relocalization of PRC2 through H3K27me3, leading to epigenetic silencing of the HOXD locus [73].

## 5. LncRNAs and Diagnosis of Cancer

Tracking the levels of lncRNAs present in the body fluids in cancerous patients can be an effective lncRNA-based diagnostic marker for such cancers. For example, lncRNA PCA3, released in the urine of prostate cancer patients, has been a sensitive and more specific marker for these patients than serum prostate-specific antigen testing, and it is also a convenient and less invasive procedure [71]. Similarly, lncRNAs, such as H19, TINCR, AOC4P, BANCR, LINC00857, and CCAT2, are detectable in body fluids and can be used to efficiently differentiate gastric cancer patients from healthy controls [71,74]. Thus, in the future, lncRNAs could be used as a very proficient and cost-effective diagnostic tool in clinical practice.

## 6. LncRNAs and Therapy Resistance in Cancer

Growing evidence suggests that lncRNAs are intimately involved in cancer therapy resistance via multiple modes of action. For example, upregulated expressions of NEAT1 and HOTAIR are responsible for therapeutic resistance in BC, OC, and various other cancer cells to chemotherapy, e.g., paclitaxel, 5-FU, cisplatin, tamoxifen, radio and endocrine therapies [75,76,77,78]. Similarly, H19, MIR2052HG, TINCR, DCST1-AS1, NONHSAT101069, and CASC2 are involved in fulvestrant, aromatase inhibitors, trastuzumab, doxorubicin, epirubicin, and paclitaxel resistance, respectively, in BC patients by way of different mechanisms [79]. The usual resistance mechanisms via lncRNAs to diverse therapeutic strategies may involve altered drug targets, increased drug efflux, maintenance of cancer stemness, immune response deregulation, and activated bypass signaling pathways. Thus, the in-depth understanding of the association of lncRNAs in resistance to therapies may benefit the clinical outcome of patients. They can also be used as therapeutic targets to tackle therapy resistance in such cancers. For example, therapeutic delivery of locked nucleic acids targeting LINK-A lncRNA has been seen in a preclinical study to improve the BC sensitivity to immune checkpoint inhibitors [79]. On the contrary, some lncRNAs have been seen to suppress therapeutic resistance. For instance, lncRNA LINC00968 sensitizes BC cells to chemotherapeutics paclitaxel and adriamycin by targeting and silencing WNT2 and inhibiting the Wnt2/β-catenin signaling pathway [80]. So, drug resistance suppressor lncRNAs can be used in combination with chemotherapeutics to enhance the effectiveness of such drugs.

## 7. Current Clinical Applications of lncRNAs

LncRNAs can be determined rapidly, efficiently, and cost-effectively in gastric juice, urine, blood serum, saliva, and tissues, making them exceedingly versatile analytes. Thus, considering the clinical need for more accurate predictive markers and early diagnosis of cancer patients, lncRNAs will be an efficient molecular tool that might aid in clinical management. Some lncRNAs have been approved by the U.S. Food and Drug Administration (FDA) as biomarkers for patients’ clinical management, indicating their importance in the clinic. For instance, the most important example of lncRNA approved by the FDA for routine clinical practice is PCA3, used for PROGENSA PCA3 urine-based molecular diagnostic testing of prostate cancer. Similarly, many more lncRNAs are on the way to FDA approval and are going through different phases of clinical trials, e.g., the lncRNA H19 is at the stage of clinical trials for glioblastoma, ovarian, bladder, and pancreatic cancer. Moreover, HOTAIR, MALAT-1, and NRCP are under pre-clinical trials for EC, prostate cancer, and OC, respectively [81,82]. Hence, lncRNAs associated with clinicopathologic characteristics of various diseases, including cancers, become more advanced as prognostic and diagnostic markers and drug resistance suppressors.

## 8. LncRNAs in Female-Oriented Cancers

The lncRNA transcripts show diverse functions in various biological processes, as they are seen to be involved in more or less every step of the life cycle of genes [2]. However, the deregulations and aberrant expressions of lncRNAs are linked with the incidence of diverse diseases, including tumorigenesis in human beings [83]. LncRNAs influence the biological behaviors of various cancers by functioning either as tumor suppressor lncRNAs or oncogenic lncRNAs [84].

For instance, SNHG3 is a novel lncRNA that has been identified to have a tumor-suppressor function during papillary thyroid carcinoma (PTC) development, and its silencing can activate tumor progression by modifying the AKT/mTOR/ERK signaling pathway. The study involved 62 patients and observed a 1.5-fold decrease in SNHG3 expression in PTC tissues of 38/62 (61.3%) patients [85]. Similarly, GAS5 works as a tumor suppressor lncRNA in HCC cells via GAS5/miR-182/ANGPTL1 axis [86]. On the contrary, a study conducted to determine the role of SNHG3 in breast cancer in a nude mouse model demonstrated that SNHG3 promotes cancer proliferation and metastasis by functioning as a miR-326 sponge [87]. The lncRNAs induce cancer invasion, proliferation, stemness, and metastasis by utilizing different mechanisms, i.e., by either promoting epithelial-to-mesenchymal transition (EMT) [88], regulating miRNAs expression profile via miRNAs sponging [89], modulating cell cycle including apoptotic pathways [90], interfering in mRNAs splicing [91], inducing resistance to different anti-cancer drugs [92], cellular metabolic reprogrammings [93], altering localization of proteins using direct binding [94], or regulating various associated metastatic signaling pathways [95].

Thus, it is likable to speculate that targeting the lncRNAs may help develop novel therapeutic strategies for cancers.

An increasing number of lncRNAs have been studied and found to be involved in cancers among females [96,97]. Several such lncRNAs involved in tumorigenesis or tumor suppression in female-oriented cancers are mentioned briefly in Figure 3. Different factors such as age, menopause status, and obesity are seen to be associated with increased cancer risks among females, and involvement of lncRNAs in such circumstances has been observed and studied by various researchers. For instance, Xu et al. observed in a case–control study involving 439 BC patients along with 439 age-matched healthy controls that the rs3787016 TT genotype (*p*-value 0.018) of SNP in lncRNAs is associated with the incidence of BC among females, especially with an enhanced risk among premenopausal females [98]. Similarly, a study presented evidence of a link between lncRNAs expression and the reproductive and obesity-related factors in the breast tissue of healthy women [99]. The same lncRNAs, LSINCT5 and GAS5, have also been extensively studied and found to be involved in breast tumors [100]. Cross-talk between lncRNAs and hormone signaling is also seen in cancer incidence among women. The ER signaling pathway promotes tumor progression and the cross-talk between such ER signaling pathway and cell cycle regulation, conducted by lncRNA MAFG-AS1 via MAFG-AS1/miR-339-5p/CDK2 axis, has also been identified in ER+ breast cancer cells, possibly promoting tamoxifen resistance [101]. In the following sections, we will review and highlight the characteristics and functions of some of the lncRNAs, i.e., HOTAIR, NEAT1, H19, MALAT1, and MEG3, frequently involved in several female-oriented cancers.

### 8.1. HOTAIR

HOX transcript antisense intergenic RNA (HOTAIR) is a trans-acting lncRNAs located between HOXC11 and HOXC12 genes on chromosome 12q13.13, with 2158 nucleotides long and consisting of six exons [102]. Early studies exposed the HOTAIR as a significant regulator of the chromatin status and a facilitator of transcriptional silencing. It can recruit PRC2 (polycomb repressive complex 2) and causes transcriptional repression by trimethylation of histone H3 lysine 27 (H3K27) complex in the HOXD locus, while at the 3′ end it can interact with LSD (the lysine-specific histone demethylase) complex of LSD1/CoREST/REST which also leads to gene silencing [103]. HOTAIR suppresses the expression of various tumor suppressor genes. It is overexpressed in a variety of primary and metastatic cancers, including female-oriented cancers (as shown in Figure 4) and several other cancers, such as hepatocellular carcinoma, pancreatic cancer, prostate cancer, lungs carcinoma, colorectal cancer, vulvar cancer, and esophageal squamous cell carcinoma (ESCC) [104,105].

In BC, overexpressed HOTAIR plays a crucial role in tumor progression by directly or indirectly modulating several molecular pathways involved in growth, malignant proliferation, invasion, self-renewal, EMT, metastatic spread, and poor prognosis, including drug resistance. For instance, the elevation of HOTAIR expression in ER+ BC cells, regulated by estradiol (E2), leads to tumor formation [106]. Similarly, in a recent study that evaluated normal and BC tissues of 15 patients, HOTAIR has been reported to promote BC development and migration through upregulation of either BCL-W (an anti-apoptotic protein) via miR-206 sequestering (*p* < 0.001) [107], SOX2 via epigenetic suppression of miR-34a [53], HMGA2 via miR-20a-5p suppression [108], ZEB1 via miR-601 sponging [109], or by regulating a broad spectrum of various other miRNAs as well. Moreover, HOTAIR can mediate the oncogenic action of c-Myc, by competing with BRCA1 (a tumor suppressor gene) [103] and boost the cancerous cells resistance to radiotherapy and various targeted drug therapies like lapatinib and imatinib [110]. In gynecological cancers, HOTAIR aberrant expression has been involved in the oncogenic progression, lymph node metastases, and poor prognosis of EC [104]. In EC, HOTAIR gene expression is seen to be increased by E2 in 23 human EC tissues compared with normal tissues, which in turn upregulates the NPM1 expression by interacting with miR-646 in EC cells leading to the metastasis of EC cells [111]. Similarly, HOTAIR can promote EC proliferation by activating the PI3K/Akt signaling pathway by binding to PTEN [112]. Furthermore, downregulation of HOTAIR expression in EC leads to overcoming progesterone resistance via epigenetic regulation of progesterone receptor isoform B [113].

In CC, enhanced HOTAIR expression can lead to progressive tumor stages, lymphatic node and lymphatic vessel metastasis, adenocarcinoma, and poor prognosis. HOTAIR can increase the possibility of cervical cell tumor development, angiogenesis, and metastasis, e.g., by upregulating the expression of MMP-9, VEGF growth factors, or EMT-related genes [77,114], targeting miRNAs like miR-23b/MAPK1 axis, or regulating the expression of BCL-2 by miR-143-3p sponging [115,116].

In epithelial OC, overexpression of HOTAIR predicts elevated tumor metastasis and poor prognosis. The regulation of EMT-related genes and specific matrix metalloproteinases (MMPs) by HOTAIR is believed to be responsible for OC metastasis [117,118]. Moreover, the hyper-expression of HOTAIR is also co-related with specific miRNAs expressions [119]. For instance, HOTAIR upholds the OC stem cells stemness by upregulating a protein-coding gene TBX3 through the miR-206/TBX3 axis [120]. HOTAIR also causes paclitaxel and cisplatin resistance by increasing the CHEK1 protein level and sponging miR-138-5p to avoid its binding to SIRT1 and EZH2, respectively [78,121].

HOTAIR has also been investigated to be involved in the proliferation, invasion, apoptosis, and migration in VSCC [122].

### 8.2. NEAT1

Nuclear Enriched Abundant Transcript 1 (NEAT1) is a lncRNA located on chromosome 11q13.1 with having two isoform transcripts, 3.7 kb long NEAT1_1 and 23 kb long NEAT1_2, that differ in their 3′ UTR region processing [76]. NEAT1 is involved in the nuclear paraspeckle formation, where it associates with various paraspeckle proteins like PSPC1, p54nrb, and SFPQ. To form an RNA–protein complex, the long transcript NEAT1_2 interacts with SFPQ/PSF and p54nrb/NONO, followed by the recruitment of NEAT1_1 and PSPC1 to this complex [123]. NEAT1 epigenetically regulates the gene expression by either modifying gene transcription and translation via recruiting or sequestering TF to or from gene promoters, modulating RNA splicing and protein stabilization via associating with RBPs, or sponging miRNAs to vary the expression of their target RNAs. Studies have demonstrated the dysregulation of NEAT1 expression during the progression of cancer, where its expression is considerably linked to tumor size, distant metastasis, TNM stage, drug resistance, and patient survival [123,124,125].

NEAT1 is overexpressed in BC cells and closely associated with cancerous cell proliferation, advanced clinical stages, lymph node metastases, chemoresistance, and poor patient prognosis [126]. A recent study involving 106 BC patients revealed that overexpressed NEAT1 (*p* < 0.05) in cancer patients as compared to healthy individuals facilitates the BC cell’s proliferation and migration by regulating the expression level of RTCB and CBX7 genes, i.e., downregulation of RTCB and upregulation CBX7, probably by binding to DNA in the nucleus [127]. Moreover, NEAT1 has also been seen to induce chemoresistance [76]. It prompts cancer proliferation, EMT, and metastasis by interacting with miRNAs. For example, it can upregulate the expression of miR-21, which in turn upregulates the RRM2 expression level and BC propagation [128], or downregulate the expression of miR-146b-5p in BC cells [129].

In EC, the expression level of NEAT1 and some positively associated genes, LEF1, MMP9, and c-myc has been upregulated with downregulation of miRNA-146b-5p via the Wnt/β-catenin signaling pathway, which can be regulated in vice versa by progesterone treatment therapy to suppress the EC [130]. Similarly, overexpressed NEAT1 has been investigated to drive the aggressive progression, invasion, migration, apoptotic suppression, and drug resistance in EC cells by facilitating TIMD4 expression via sponging miR-202-3p [131], by regulating the miR-144-3p/EZH2 axis [132], and by elevating the expression of oncogene STAT3 and other TM related genes including ROCK1, MEF2D, VEGF-A, WNT7A, KPNA4, and PDE4B through sponging the tumor suppressor miR-361 [133].

The expression of NEAT1 has also been upregulated in CC cells and positively correlated with lymph node metastasis and the TNM stage. As such, overexpression of NEAT1 can accelerate the proliferation and migration of CC by regulating the miR-124/NF-κB pathway [134], by regulating the miR-889-3p/E2F7 axis via activation of the PI3K/AKT pathway [135], or by sponging various other miRNAs [136,137]. A recent study demonstrated that overexpressed NEAT1 could suppress the expression of miR-361, leading to the elevated levels of an EMT key activator HSP90 and consequently enhanced sphere formation and EMT in CC cells [138].

In OC, NEAT1 promotes cancer cell proliferation, invasion, migration, EMT, and angiogenesis by regulating the expression of a wide variety of miRNAs and associated pathways. For example, upregulation of the FGF9 pathway by sponging miR-365 [139], regulation of TJP3 expression by interacting with and sponging miR-1321 [140], alteration of cancer proliferation, apoptosis and colony formation by regulation of miR-4500/BZW1 axis [141], and by sponging many other miRNAs [142]. Moreover, in a study involving 18 cisplatin-sensitive and 19 cisplatin-resistant OC patients, overexpressed NEAT1 was reported to induce cisplatin resistance (*p* = 0.031) in OC cells via the regulation of miR-770-5p and PARP1 [143].

Similarly, NEAT1 has been documented in the development and progression of VSCC [144]. A detailed model of the mechanisms involved in cancer progression by NEAT1 is presented in Figure 5.

### 8.3. H19

The lncRNA H19 consists of a 2.3-kb long transcript and sites on the human gene cluster of H19/IGF2 on chromosome 11p15.5. It is encoded by the gene H19, one of the imprinting genes with maternal expression, fully capped, polyadenylated, spliced, and comprises five exons with four introns [145,146]. H19 is considered bi-functional RNA, as it works both as a lncRNA, a precursor for miR-675, and is involved in promoting normal biological processes like angiogenesis, inflammation, neurogenesis, apoptosis, and cell death [147]. It is an oncofetal lncRNA widely expressed in the embryo and downregulated at birth; however, it is replenished in different tumors. Furthermore, the H19 single nucleotide polymorphism (SNP), such as rs3741219, rs3024270, rs217727, rs2839698, rs2107425, and rs2735971 in various ethnic populations is seen to be associated with the susceptibility of multiple cancers like pancreatic cancer, colorectal cancer, lung cancer, OSCC, glioma, BC, and gynecological cancers [148,149,150]. However, there is always a debate about whether H19 works as an oncogenic factor or a tumor suppressor [151]. H19 regulates the gene expression by either recruitment of epigenetic RFs to the chromatin surface and regulating the gene expression by way of methylation or by regulating the two miRNAs (i.e., miR-675-3p and miR-675-5p), which are derived from exon 1 of H19 and in turn regulate the gene expression, or by interacting with and sponging several other miRNAs [152].

In BC, the aberrant expression of H19 is associated with the proliferation and progression of the tumor by diverse underlying molecular mechanisms, including interaction with c-myc, encoding microRNA-675, and competition for the regulation of endogenous RNAs [153,154]. 43 BC patients evaluated at Zhejiang University (Hangzhou, China) showed that the H19/let-7/Lin28 ceRNA pathway is involved in cellular proliferation, autophagy, and EMT in BC cells. Such H19/let-7/Lin28 loop inhibits autophagy and promotes EMT in breast cancer cells by exhibiting a significant positive relationship between H19 (*p* = 0.0317) and Lin28 (*p* = 0.0128) expression [155]. Furthermore, H19 has been observed to promote cancer cell invasion, EMT process, and lymph node metastases by enhancing the expression of its target gene TNFAIP8 via antagonizing tumor suppressor p53 [156].

In EC, the overexpressed lncRNA H19 promoted cancer cell proliferation considering fresh EC tissues of 43 cases from July 2010 to July 2012 via upregulating the expression of the HOXA10 gene by competitively targeting and downregulating miR-612 expression. It can also promote EC aggressiveness by modulating the EMT process [157,158], although it is not expressed in the normal endometrial epithelium [157]. H19 binds to let-7 and downregulates its expression, leading to enhanced expression levels of let-7 targets (Imp3, c-myc, and HMGA2) and causing enhanced EC cell invasion and migration [159]. Moreover, Zhu et al. analyzed 56 pairs of CC and adjacent normal tissues collected from CC patients (*p* < 0.05) and showed the involvement of H19 in tumor formation of EC based on its ability to regulate the miR-20b-5p/AXL/HIF-1α signaling pathway [160].

Downregulated H19 has been involved in CC proliferation, invasion, metastasis, and radioresistance [161]. In previous studies, loss of imprinting (LOI) of H19 has also been associated with the incidence of various diseases, including CC and EC [152,162]. Ou et al. reported the negative correlation between H19 and miR-138-5p expression in CC patients since downregulation of miR-138-5p expression promotes tumor development and proliferation via upregulation of its target SIRT1 [163].

In OC, H19 acts as an oncogenic lncRNA because overexpressed H19 causes the inhibition of OC cell’s apoptosis by regulating specific apoptosis-related proteins [97]. Like in CC, LOI of H19 genes may also be involved in the proliferation of OC cells [164]. A study on TGF-β induced EMT in OC cells demonstrated that H19 prompts such EMT process by competitively binding and sponging miR-370-3p because overexpressed miR-370-3p reasons the suppression of TGF-β-induced EMT in OC cells [165]. H19 has also been responsible for cisplatin resistance in OVCAR3 cells, and Sajadpoor et al. showed that the downregulation of H19 by valproic acid could endorse the cisplatin sensitivity and apoptosis in OC A2780 cells [166]. Moreover, polymorphisms in IGF2/H19 gene locus are presumably associated with platinum resistance in OC. Interestingly, the H19-rs4244809 GG genotype is seen to be associated with a reduced risk of platinum resistance, while rs3842761 and rs4244809 are connected with a significant risk of platinum resistance in FIGO stage III-IV [167].

Abnormal expression of H19 has also been linked with GTN, especially choriocarcinoma [168]. A study showed that downregulation or knockdown of H19 expression by lentiviral vectors expressing H19-specific siRNA obstructed the proliferation of human choriocarcinoma cell line JAR [169]. Furthermore, S. Yu et al. studied that H19 is also related to the chemo-resistance mechanisms in choriocarcinoma cells, and they found that, after knock out of H19 from JEG-3/5-FU and JEG-3/MTX cells, the drug resistance index was diminished, leading to significantly reduced cancer proliferation, invasion, and migration with simultaneously increased apoptosis [170]. For a hypothetical model on the role of H19, see Figure 6 below.

### 8.4. MALAT1

Metastasis Associated Lung Adenocarcinoma Transcript 1 (MALAT1), also known as NEAT2, is ~8 knt long in humans, with its gene located on chromosome 11q13.1. It is nuclear retained and highly conserved, exhibiting more than 80% conservation at the transcript’s 3′ end [171]. It also displays an unusual 3′ end processing and significant evolutionary conserved secondary and tertiary structural features [172]. It is richly expressed in tissues and cells and regulates gene expression in a context-dependent manner, both at the transcriptional and post-transcriptional levels [173]. Accumulated evidence confirmed that it shows a low protein-coding potential by two independent coding potential calculating algorithms, i.e., CPAT and CPC2 [174]. Moreover, MALAT1 is localized at nuclear speckles being enriched at the periphery, although it is not involved in nuclear speckles formation [175].

It has been evidenced to regulate gene transcription, directly or indirectly, either by binding with histone modification enzymes or transcription factors. Post-transcriptionally, it regulates the mRNAs and protein expression by competitively binding miRNAs and acting as a sponge to sequester miRNAs [176,177,178,179]. MALAT1 has been seen to be associated with different diseases, including cancer. Initially, its elevated expression was identified in primary lung cancer cells with a high tendency of metastasis [180]. Since that overexpressed MALAT1 has been reported to be linked with a wide variety of lymphoid or solid tumors with high tumor progression and metastasis propensity and 1.5–10 fold relative upregulation based on type and stage of cancer [172,181,182,183,184]. Moreover, single nucleotide polymorphism in MALAT1, e.g., rs619586 A > G polymorphism, has also been witnessed to be linked with elevated cancer risks [185]. On the contrary, some recent studies have postulated the downregulated expression of MALAT1 in human breast and colorectal cancer, where its decreased expression is associated with lessened patient survival [176,186].

In BC patients, MALAT1 is considered a possible indicator for early prognosis and diagnosis, as its expression is seen to be upregulated in such patients with downregulation of its expression in patients receiving breast-conserving surgery in combination with chemotherapy [187,188]. Stone et al. postulated that hypoxia in BC cells could also mediate the upregulation of MALAT1 by chromatin looping [173]. Such overexpressed MALAT1 can promote BC proliferation and progression by repressing various RNA molecules. For example, it can competitively bind with miR-1 and thereby affect the expression of CDC42 [189], downregulate hsa-miR-448, lead to aberrant expression of KDM5B [190], and regulate the expression of miR-143-3p and its putative target, RALGAPA2 [191]. Huang et al. detected the MALAT1 expression levels in tissue samples collected from 20 BC patients and 20 healthy controls and found significantly higher expression levels in the former collective (*p* < 0.05). They concluded that this increased expression level of MALAT1 also induces angiogenesis in BC cells by downregulating the expression of miR-145 and upregulating VEGF expression [192]. Furthermore, MALAT1 can interact with numerous pathway target genes closely linked with tumor proliferation and metastasis. MALAT1 has expedited a pro-metastatic state in BC by trans-regulating the EEF1A1 epigenetic pathway after binding to its promoter regulatory element [193]. It can promote the invasion and proliferation of BC cells by regulating the XBP1-HIF-1α pathway and HER-2 pathway in MDA-MB-23 and MDA-MD-435 cell lines, respectively [194], and assist in promoting trastuzumab resistance in HER2 overexpressing BC cells via FOXO1 and PI3/Akt pathway [195].

Studies have shown that MALAT1 can promote the proliferation, invasion, and metastasis of EC cells [196] with MALAT1 gene SNP, such as rs664589C > G polymorphism reported to considerably increase the risk of EC in females [197]. MALAT1 can repress the expression and function of miR-200c by competitively binding and sponging it and thus regulating the function of TGFβ in EC cells [198]. The PCDH10–Wnt/b-catenin–MALAT1 regulatory axis has also been involved in the development and proliferation of EC, in vivo and in vitro [199]. Moreover, Y. Shen et al. established that on the treatment of EC cell lines with MEK inhibitor RG7420, MALAT1 can decrease up to 6.13 times with upregulation of tumor suppressor miR-129-5p and downregulation of TAK1, leading to decreased metastasis and increased apoptosis of EC cells [200].

In CC, the expression of MALAT1 is known to be significantly augmented in cancer cells and tissues [172], most probably by IL-6/STAT3 and HPV18 E6/E7 mediated signaling pathways [184]. MALAT1 is seen to be structurally upregulated in CC cells. A recent study demonstrated that it undergoes 18 different structural rearrangements in CC-derived HeLa cells leading to the effect of more than 50 validated miRNA-binding sites by such putative secondary structure [201]. Thus, it prompts cancer proliferation and EMT in CC cells, for example, by sponging of miR-145, which was previously reported to suppress the tumor progression in CC cells by the regulation of Cyclin D1and CDKs [202], sponging of miR-202-3p and leading to upregulation of periostin expression [203], and sponging of miR-429 [204]. Furthermore, by sponging miR-625-5p, overexpressed MALAT1 has been characterized to weaken the inhibitory effect of miR-625-5p on NF-κB signaling in CC cell growth [205]. MALAT1 also promotes chemoresistance in CC patients, e.g., induction of cisplatin resistance by regulating BRWD1 and PI3K/AKT pathway [206].

X. Wu et al. have shown the involvement of MALAT1 in OC cell’s stemness, non-adherent spheres formation, and cisplatin resistance via enhancing YAP expression and activity by inhibition of its nuclear-cytoplasm translocation [207]. Likewise, Bai et al. indicated the association of ABCC1 and Notch1 signaling pathway for cisplatin resistance in OC because of MALAT1 by experimenting with 20 paired tumor tissue samples taken from OC patients and adjacent normal tissue samples [208]. By negatively regulating the expression of miR-143-3p and miR-503-5p via sponge-like action, MALAT1 promotes progression and proliferation of OC through activation of CMPK and JAK2-STAT3 pathways, respectively [209,210]. Moreover, it also functions as an oncogenic lncRNA by sponging miR-200c in OC cells [182]. MALAT1 facilitates the metastasis of OC by promoting RBFOX2-mediated alternative splicing and EMT and regulating the expression of pro-apoptotic tumor suppressor gene KIF1B [211]. Similarly, a study involving sixty OC patients as well as OC cell lines with a *p*-value of *p* < 0.05 showed that the overexpression of MALAT1 prompts cell proliferation, metastasis, and cell cycle progression in the S phase with inhibition of cell apoptosis via activation of β-catenin, DVL2, cyclin D1, and Wnt/β-catenin signaling pathway in OC cells with decreased expression of GSK-3β [212].

MALAT1 may also represent a possible diagnostic biomarker for VSCCs and choriocarcinoma [213,214], but the studies about their incidence mechanisms are limited. However, Shi et al. showed that MALAT1 could promote choriocarcinoma tumor growth via regulation of miR-218-mediated Fbxw8 [215]. The detailed role of MALAT1 in female-oriented cancers is modeled below in Figure 7.

### 8.5. MEG3

A maternally Expressed Gene 3 (MEG3) is a lncRNA encoded by the MEG3 gene and is 1.6 kb long. It is located in human chromosome 14q32.3, inside the imprinted DLK1-MEG3 locus. The MEG3 gene is a maternally imprinted gene, is 35 kb long, and contains ten exons. Multiple factors are associated with the regulation of MEG3 gene expression, including cyclic adenosine monophosphate, DNA methyltransferase family, and nuclear factor-κB (NF-κB) [216,217]. MEG3 localizes both in the cytoplasm and nucleus [218] and is abundantly expressed in many tissues where it plays a prominent role in development and growth. At the same time, the loss of imprinting of MEG3 may lead to moderate to severe developmental disorders [219]. Moreover, SNP within the MEG3 intron may increase, among others, the susceptibility of breast cancer, oral squamous cell carcinoma, and type 1 diabetes [220,221,222]. The expression of MEG3 is downregulated in numerous human primary cancers or cancer cell lines where it functions as an antitumor component or a tumor suppressor, such as glioma, liver, breast, cervical, lung, ovarian, osteosarcoma, colorectal, bladder, prostate, and gastric cancer cells [223]. Thus, restoring the MEG3 expression can inhibit the cancer cell’s proliferation and prompt their apoptosis [224]. MEG3 might function by regulating the foremost tumor suppressor genes p53 and Rb, controlling miRNAs, or inhibiting angiogenesis-related factors. However, dysregulation of MEG3 expression may lead to the development and proliferation of cancer, suggesting a potential biomarker and therapeutic target in human cancers [223,225].

In BC cells, MEG3 inhibits cellular proliferation and induces apoptosis by activating the endoplasmic reticulum stress or inducing p53 activation via the NF-κB signaling pathway [226]. The SNP, such as GG of MEG3 rs3087918, has also been associated with a decreased risk of BC, while MEG3 haplotype TCG possibly increases the risk of BC initiation [225]. Bayarmaa et al. postulated that MEG3 polymorphism seems to be associated with chemotherapy response and toxicity of cisplatin and paclitaxel in BC patients [227]. Moreover, Mingzhi Zhu et al. recruited 31 BC patients (*p* < 0.05) and found that MEG3 suppresses BC cells growth, migration, and invasion and induces paclitaxel resistance as well as cancer cells apoptosis via modulating the expression of miR-4513 and PBLD [228]. However, Ali et al. showed that MEG3 rs7158663 is associated with increased cancer susceptibility, even with higher TNM staging and tumor size > 5 cm, via altering its gene expression level [229] and hyper-methylated MEG3 induces chemoresistance in BC cells [230].

Overexpressed MEG3 significantly reduces cancer proliferation and metastasis and can induce apoptosis of EC cells [159] either by regulating the Notch1 signaling pathway [231,232] or by downregulating the expression of PI3K protein and its downstream genes, including BCL-XL, VEGF-A, P70S6K, and Mtor [233]. Furthermore, Xu et al. evaluated a total of 65 human EC tissues and 18 normal samples and showed that MEG3 downregulates the miR-216a expression, leading to increased expression of tumor suppressor PD-L1. Thus, it inhibits the EC cell migration and invasion [234].

In CC cells, MEG3 serves as a prognostic indicator and diagnostic marker, and its expression is seen to be associated with HR-HPV infection, lymph node metastasis, tumor size, and FIGO staging [235]. At the same time, overexpressed MEG3 shows the potential to inhibit cancerous cells proliferation and induce apoptosis [236]. It has also been shown to suppress CC cells growth by downregulation of miR-21-5p levels in CC cell lines, by upregulation of the expression of SCT1 glycoprotein by sponging miR-7-5p, leading to the endoplasmic reticulum (ER) stress-mediated apoptosis of CC cells, or by ubiquitination of P-STAT3 [237,238,239]. Moreover, a recent study showed that lidocaine could inhibit the proliferation of CC by increasing the expression of MEG3 in HeLa cells. Such overexpressed MEG3 downregulates the expression of miR-421, leading to upregulation of BTG1 expression, which is negatively correlated with the expression of miR-421 [240].

The decreased expression of MEG3 is also considered a hallmark for tumor progression in OC [241]. J. Wang et al. showed that MEG3 inhibits cancer cell proliferation and induces apoptosis by regulating the expression of downstream tumor suppressor gene PTEN in OC [242]. Likewise, the overexpressed MEG3 inhibits cell proliferation and EMT in OC cells by sponging miR-205-5p via regulation of miR-219a-5p/EGFR axis and by regulating the expression of LAMA4 via miR-30e-3p sponging [243,244,245]. However, a recent study demonstrated that anisomycin inhibited proliferation, invasion, and angiogenesis in OC cells via inhibition of the Notch1 pathway by attenuating the molecular sponge effect of the MEG3/miR-421/PDGFRA axis [246].

MEG3 also represses the proliferation, migration, invasion and induces apoptosis of BeWo and JEG-3 of human choriocarcinoma cells through upregulating miR-211, leading to inhibition of PI3K/AKT and AMPK pathways [247]. Moreover, a multidrug resistance-reversing agent, Schisandrin A, has been reported to repress the tumor growth in choriocarcinoma cells by upregulating MEG3 expression and downregulating PI3K/AKT/NF-κB signal cascade [248]. We summarized the detailed response of MEG3 to female-oriented cancers in Figure 8.

Several LncRNAs involved in proliferation, invasion, apoptosis, migration, metastasis, and drug resistance in female-orieted cancers are briefly discussed in Table 1, Table 2, Table 3, Table 4, Table 5 and Table 6.

## 9. Closing Remarks and Future Directions

The status of lncRNAs will change day by day with more and more knowledge and understanding in molecular biology and oncology. By understanding the underlying mechanisms and functions of lncRNAs in cancer cells, efficient therapeutic approaches could be determined. Furthermore, it is expected that lncRNAs might be used as potential biomarkers for early diagnosis or prognosis of cancer because of their aberrant changes during cancer progression, such as the use of lncRNA PCA3 for early diagnosis of prostate cancer with high sensitivity and specificity [297]. The actual application of lncRNAs as potential biomarkers and targets for diagnosis and therapy has broad prospects for future cancer treatment and for modifying therapy according to the needs of individual patients. However, significant development and research efforts are still needed to determine the utilization of lncRNAs-based technologies in clinical utility.

## Figures and Tables

**Figure 1 cancers-13-06102-f001:**
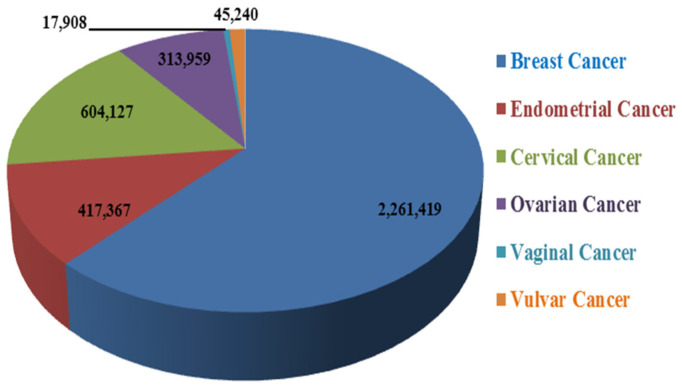
Worldwide incidence of female-oriented cancers according to WHO calculations.

**Figure 2 cancers-13-06102-f002:**
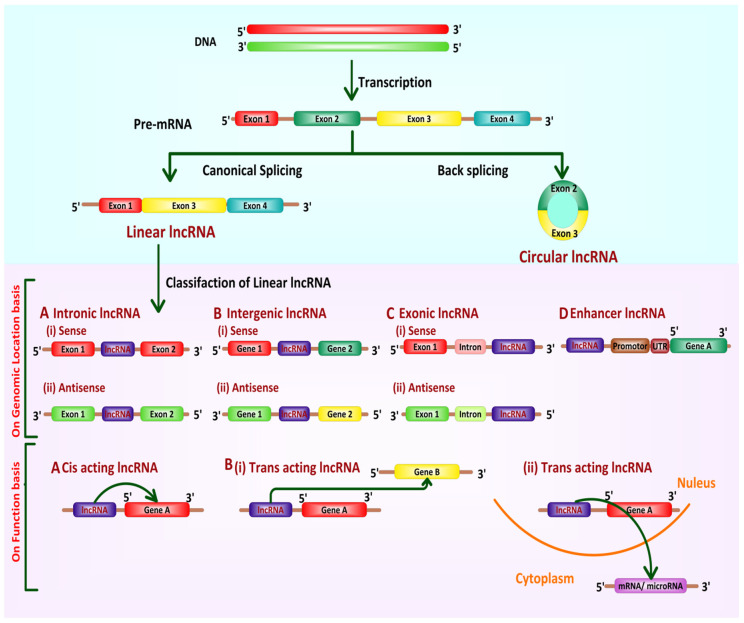
Biogenesis and classification of lncRNAs. Canonical splicing produces linear lncRNA while circular lncRNA is made via back splicing. Linear lncRNAs are further categorized based on their genomic location, direction of transcription (A= (i) Sense and (ii) Antisense intronic lncRNA, B = (i) Sense and (ii) Antisense intergenic lncRNA, C = (i) Sense and (ii) Antisense exonic lncRNA, and D = Enhancer lncRNA) and on their functions (A = Cis-acting lncRNA and B = (i) & (ii) Trans-acting lncRNA).

**Figure 3 cancers-13-06102-f003:**
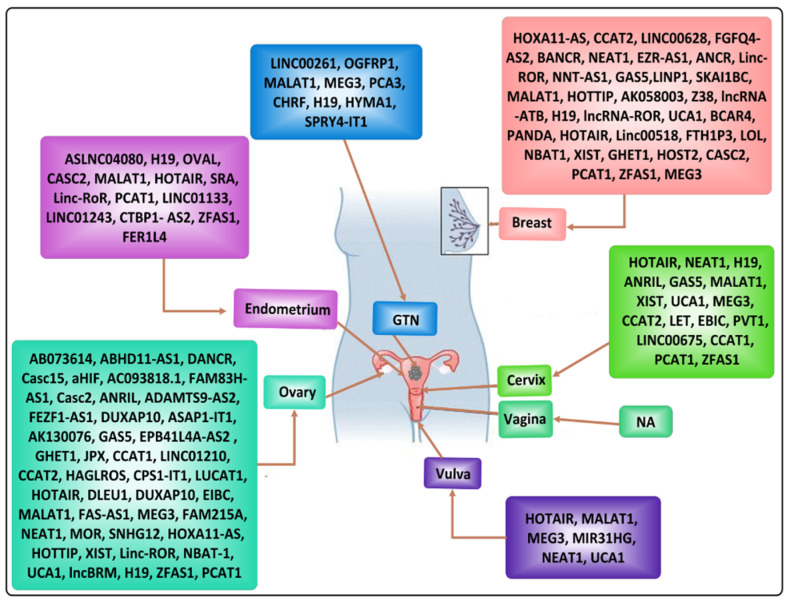
Most common lncRNAs associated with female-oriented cancers.

**Figure 4 cancers-13-06102-f004:**
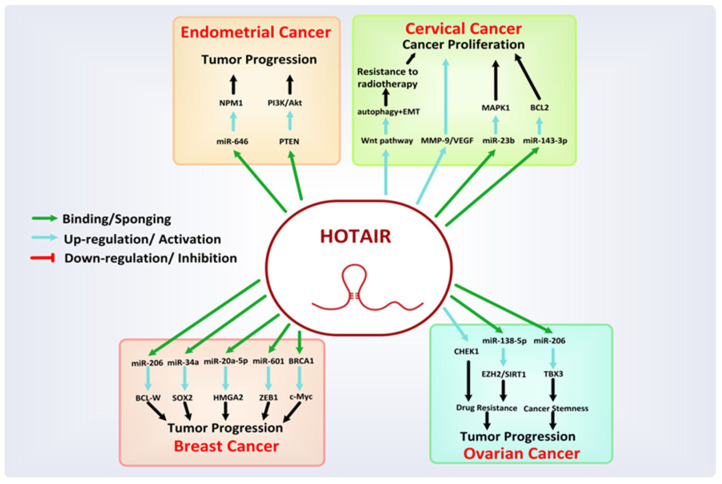
Mechanism associated with the oncogenic role of HOTAIR in female-oriented cancers.

**Figure 5 cancers-13-06102-f005:**
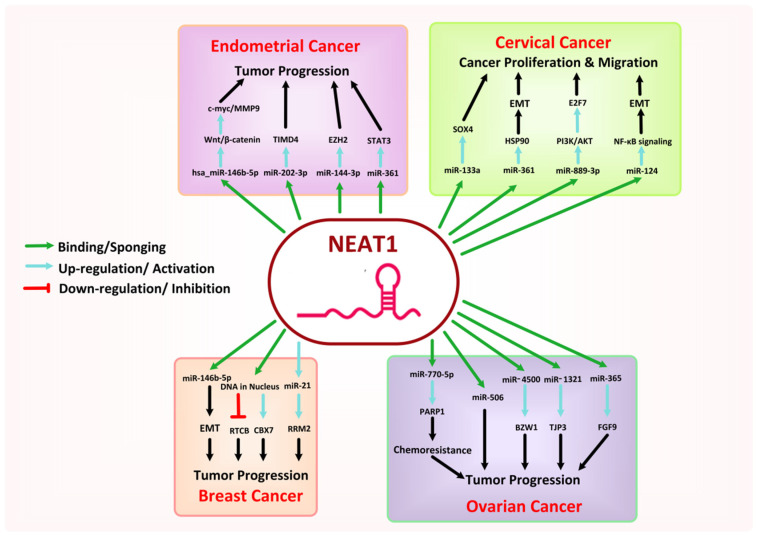
Mechanism associated with the oncogenic role of NEAT1 in female-oriented cancers.

**Figure 6 cancers-13-06102-f006:**
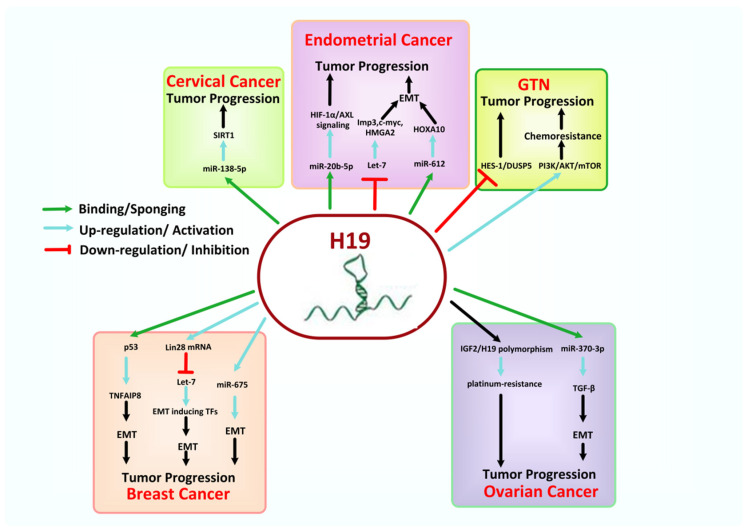
Mechanism associated with the oncogenic role of H19 in female-oriented cancers.

**Figure 7 cancers-13-06102-f007:**
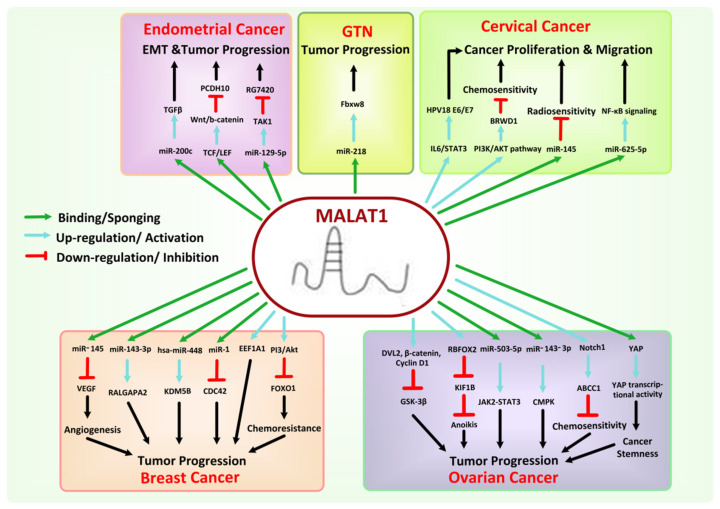
Mechanism associated with the oncogenic role of MALAT1 in female-oriented cancers.

**Figure 8 cancers-13-06102-f008:**
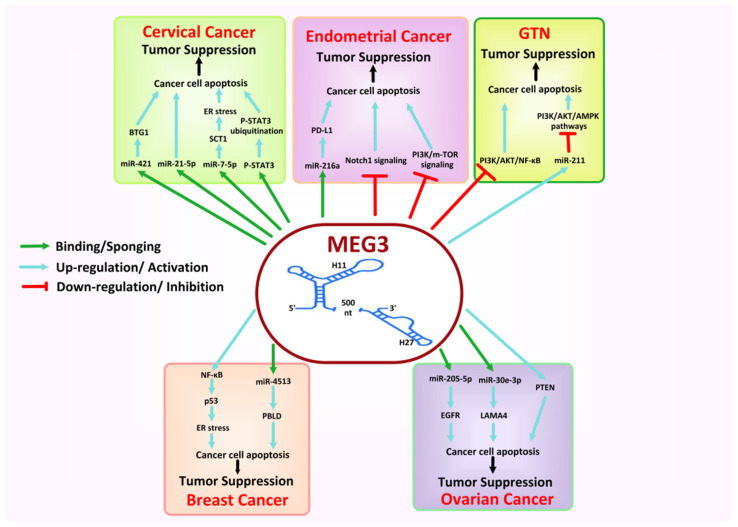
Mechanisms associated with the tumor suppressor of MEG3 in female-oriented cancers.

**Table 1 cancers-13-06102-t001:** LncRNAs involved in proliferation, invasion, apoptosis, migration, metastasis, and drug resistance in breast cancers.

LncRNAs	Locus	Status	Target/Function	References
HOXA11-AS	7p15.2	Oncogenic	EMT process	[22]
CCAT2	8q24.21	Oncogenic	OCT4-PG1, Wnt/B-catenin, Notch signaling pathway	[249]
HOTTIP	7p15.2	Oncogenic	miR-615−3p/HMGB3, E-cadherin, N-cadherin, Snail, twist, PI3K/AKT, Wnt/β-catenin pathway	[250]
NEAT1	11q13.1	Oncogenic	RTCB, CBX7, EMT process, miR-21/RRM2, miR-146b-5p	[128,129]
LUCAT1	5q14.3	Oncogenic	miR-5702, miR-7-5p, SOX2	[251]
Linc-ROR	18q21.31	Oncogenic	EMT process via miR-205	[96]
lncRNA-ATB	chr 13,	Oncogenic	EMT process via targeting miR-141-3p	[252]
LINP1	10p14	Oncogenic	EMT process by anti-metastatic effects of P53	[96]
Z38	3q12.1	Oncogenic	N/A, silencing promotes apoptosis in breast cancer	[253]
SKAI1BC	-	Oncogenic	KAI1/CD82 metastasis suppressor gene	[96]
NNT-AS1	5p12	Oncogenic	miR-142-3p/ZEB1 axis	[254]
AK058003	10q22	Oncogenic	gamma-synuclein gene (SNCG)	[253]
LINC00628	1q32.1	Tumor suppressor	BCL-2/BAX/Caspase-3 signaling pathway	[255]
ANCR	4q12	Tumor suppressor	EMT via E2H2	[96,256]
MALAT1	11q13.1	Oncogenic	miR-1/CDC42, miR-143-3p/RALGAPA2, EEF1A1, XBP1-HIF-1α, HER-2 pathway	[191,193,194]
GAS5	1q25.1	Tumor suppressor	miR-23a, PTEN, miR-21	[257,258]
BANCR	9q21.11	Oncogenic	MMPs, EMT, BAX, Caspase 3, PARP	[96]
H19	11p15.5	Oncogenic	c-myc, miR-675, Let-7/Lin28, EMT via TNFAIP8/p53	[155,156]
UCA1	19p13.12	Oncogenic	SATB1, ARID1A/CEBPα, EMT by TGF-β, p27 (Kip1), miR-122-5p, Wnt/β-catenin pathway	[259]
BCAR4	16p13.13	Oncogenic	Wnt/β-catenin, YAP/Hh signaling pathways, ERBB2, EMT via mTOR signaling	[260]
HOTAIR	12q13.13	Oncogenic	miR-206/BCL-W, miR-34a/SOX2, c-Myc/BRCA1,many othr miRNA	[53,103,107]
FAM83H-AS1	8q24.3	Oncogenic	miR-136-5p/MTDH axis	[261]
NBAT1	6p22. 3	Tumor suppressor	DKK1, EZH2, PRC2	[255]
XIST	Xq13	Tumor suppressor	miR-155/CDX1 axis, c-Met pathway	[255]
GHET1	7q36.1	Oncogenic	N-cadherin, Vimentin, E-cadherin	[255]
PCAT1	8q24	Oncogenic	HIF-1a/RACK1 pathway	[262]
ZFAS1	20q13.13	Oncogenic	miR-589, MMP9, MMP2, BCL-2, Caspase-3, PTEN, BAX, N-cadherin, E-cadherin, Vimentin PI3K/AKT pathway	[263]
HOST2	10q23.1	Oncogenic	miR Let-7b pathway	[255,264]
CASC2	10q26	Tumor suppressor	miR-96-5p/SYVN1 pathway	[255]
MEG3	14q32.3	Tumor suppressor	NF-Κb/p53 pathway, miR-4513/PBLD	[226,228]

**Table 2 cancers-13-06102-t002:** LncRNAs involved in proliferation, invasion, apoptosis, migration, metastasis, and drug resistance in endometrial cancers.

LncRNAs	Locus	Status	Target/Function	References
HOTTIP	7p15.2	Oncogenic	PI3K/AKT pathway	[250]
NEAT1	11q13.1	Oncogenic	Wnt/β-catenin signaling, miR-202-3p/TIMD4, miR-144-3p/EZH2, miR-361/STAT3	[130,131,133]
ASlnc04080	-	Oncogenic	Unknown	[265]
H19	11p15.5	Oncogenic	EMT via Let-7 targets Imp3, c-myc, HMGA2, miR-20b-5p/AXL/HIF-1α	[159,160]
HOXB-AS1	_	Oncogenic	miR-149-3p/Wnt10b, c-Myc, β-catenin, cyclinD1	[266]
BANCR	9q21.11	Oncogenic	MMP1/2, MAPK, MEK/ERK signaling	[159]
UCA1	19p13.12	Oncogenic	AMOTp130, YAP, Hippo-YAP, miR-143, FOSL2	[267]
PCGEM1	2q32	Oncogenic	miR-129/STAT3	[268]
MALAT1	11q13.1	Oncogenic	miR-200c/TGFβ, PCDH10–Wnt/b-catenin axis, RG7420, miR-129-5p/TAK1	[198,199,200]
MEG3	14q32.3	Tumor suppressor	Notch1, PI3K, BCL-XL, VEGF-A, P70S6K, mTOR	[231,233]
HOTAIR	12q13.13	Oncogenic	miR-646/NPM1, PTEN, PI3K/Akt signaling	[111,112]
CCAT2	8q24.21	Oncogenic	miR-216b/PI3K/AKT pathway, BCL-2	[265]
SRA	5q31.3	Oncogenic	Wnt/β-catenin, EIF4E-BP1	[265]
Linc-RoR	18q21.31	Oncogenic	miR-145, PI3K-Akt pathway	[269]
PCAT1	8q24	Oncogenic	BCL-2, vimentin, N-cadherin, E-cadherin	[270]
DLEU1	13q14.3	Oncogenic	miR-490, BAX, N-cadherin, E-cadherin, Snail, CASP-3, vimentin, SP1, PI3K, mTOR, AKT1, p70S6K, GSK3B, STAT3, BCL--2, BCL-xl,	[265]
TUG1	22q12.2	Oncogenic	VEGF-A, miR-34a, miR-299	[159]
DCST1-AS1	_	Oncogenic	miR-92a-3p/Notch1	[271]
ZFAS1	20q13.13	Oncogenic	CDK4, Cyclin-D1, Ecadherin, Ncadherin, EMT	[263]
GAS5	1q25.1	Tumor suppressor	P27/PTEN, miR-103/PTEN, miR-222-3p	[272]
FER1L4	Chr. 20	Tumor suppressor	PTEN, AKT	[159]
SNHG8	Chr. 4	Oncogenic	miR-152/c-MET	[265]

**Table 3 cancers-13-06102-t003:** LncRNAs involved in proliferation, invasion, apoptosis, migration, metastasis, and drug resistance in cervical cancers.

LncRNAs	Locus	Status	Target/Function	References
HOTAIR	12q13.13	Oncogenic	MMP-9, VEGF, EMT-related genes, miR-23b/MAPK1, miR-143-3p/BCL-2 axis	[77,114,115,116]
NNT-AS1	5p12	Oncogenic	Wnt/β–catenin pathway, miR-186/HMGB1 axis	[254]
ANRIL	9p21.3	Oncogenic	p15, miR-186, PI3K/Akt pathway	[273]
BCAR4	16p13.13	Oncogenic	EMT process	[260]
H19	11p15.5	Oncogenic	miR-138-5p	[163]
GAS5	1q25.1	Tumor suppressor	Akt, miR-106b, IER3	[257,258]
SNHG20	17q25.2	Oncogenic	miR-140-5p/ADAM10 axis	[274]
MALAT1	11q13.1	Oncogenic	IL-6/STAT3, HPV18 E6/E7, PI3K/AKT signaling pathways, miR-145/Cyclin D1, miR-625-5p/NF-κB	[184,202,205,206]
XIST	Xq13	Oncogenic	miR-889-3p/SIX1 axis, miR-23a-3p/LGR4 miR-30b-5p, miR-30c-5p, miR-30e-5p I ADAM9	[275,276]
UCA1	19p13.12	Oncogenic	VEGF, miR-206	[267]
LET	_	Tumor suppressor	Unknown	[273]
MEG3	14q32.3	Tumor suppressor	miR-21-5p, miR-7-5p/SCT1, miR-421/BTG1, P-STAT3	[237,238,239,240]
CCAT2	8q24.21	Oncogenic	TCF7L2, MYC, miR-17-5p, miR20a, Wnt/β-catenin signaling pathway	[258]
SBF2-AS1	11p15.1	Oncogenic	miR-361-5p/FOXM1 axis	[277]
EBIC	16q	Oncogenic	E-cadherin/EZH2,	[258,273]
LUCAT1	5q14.3	Oncogenic	MTA1, miR-181a, miR-199b-5p	[251]
PVT1	8q24	Oncogenic	miR-200b/EZH2, miR-128-3p, miR-424	[258,277]
CCHE1	10q21.1	Oncogenic	PCNA	[273]
TUG1	22q12.2	Oncogenic	miR-138-5p/SIRT1, Wnt/β-catenin signaling pathway	[278]
NEAT1	11q13.1	Oncogenic	miR-124/NF-κB, miR-889-3p/E2F7/PI3K/AKT, miR-361/HSP90	[134,135,138]
PCAT1	8q24	Oncogenic	Unknown	[270]
LncRNA-ATB	chr 13, 14 and 22	Oncogenic	miR-144/ITGA6 axis	[252]
SPRY4-IT1	_	Oncogenic	MiR-101-3p, E-cadherin, vimentin, ZEB1, EMT	[236]
ZFAS1	20q13.13	Oncogenic	Unknown	[263]

**Table 4 cancers-13-06102-t004:** LncRNAs involved in proliferation, invasion, apoptosis, migration, metastasis, and drug resistance in ovarian cancers.

LncRNAs	Locus	Status	Target/Function	References
HOTTIP	7p15.2	Oncogenic	Wnt/β-catenin, STAT3 signaling pathways, IL-6/PD-L1, c-jun	[250]
NEAT1	11q13.1	Oncogenic	EMT via miR-365/FGF9, miR-1321/TJP3, miR-4500/BZW1 axis, miR-770-5p/PARP1	[139,140,141,143]
HOXA11-AS	7p15.2	Tumor suppressor	Unknown	[22]
PVT1	8q24	Oncogenic	miRNA133a, miR-140, s TGF-β1, p-SMAD4, CASPASE-3	[264]
AB073614	3q24	Oncogenic	p-Akt, PTEN, PI3K/Akt, ERK pathways, BCL-2, BAK, BAX, N-cadherin, vimentin, MMP2, EMT	[279]
ABHD11-AS1	7 q11. 23.	Oncogenic	RhoC/PI3K/Akt signaling, RhoC/P70s6k, RhoC/BCL-xL	[279]
DANCR	4q12	Oncogenic	IGF2	[256,280]
FAS-AS1	10q23.31	Oncogenic	Unknown	[281]
aHIF	_	Oncogenic	Unknown	[282]
FAM83H-AS1	8q24.3	Oncogenic	HuR protein	[261,282]
HOST2	10q23.1	Oncogenic	miRNA let-7	[264]
ADAMTS9-AS2	3p14. 1	Tumor suppressor	miR-182-5p/FOXF2 signaling pathway	[279]
CASC2	10q26	Tumor suppressor	EIF4A3, PI3K/AKT/mTOR pathway, NF-κB signaling	[282,283]
ANRIL	9p21.3	Oncogenic	let-7a, HMGA2, MMP3, MET, cyclin D1-CDK4/6	[264]
FEZF1-AS1	7q31.32	Oncogenic	miR-130a-5p/SOX4 axis.	[284]
DUXAP10	14q11.2	Oncogenic	VEGF, MMP-9, E-cadherin, B-catenin, Snail, vimentin, Twist	[97,285]
ASAP1-IT1	_	Tumor suppressor	Hippo/YAP signaling	[286]
GAS5	1q25.1	Tumor suppressor	miR-196a-5p	[257]
EPB41L4A-AS2	_	Tumor suppressor	microRNA-103a/RUNX1T1	[287]
GHET1	7q36.1	Oncogenic	HIF1a/VEGF	[282]
JPX	_	Oncogenic	PI3K/Akt/mTOR pathway	[288]
CCAT1	8q24.21	Oncogenic	miR-490-3p, miR-1290, miR-3679, TGFβR1	[264]
CCAT2	8q24.21	Oncogenic	Wnt/beta-catenin pathway miR-424	[97,264]
HAGLROS	2q31.1	Oncogenic	miR-100/mTOR, miR-100/ZNRF2	[289]
CPS1-IT1	_	Tumor Suppressor	BAX, caspase-9, BCL-2	[279]
LUCAT1	5q14.3	Oncogenic	miR-612/HOXA13 axis, miR-612/HOXA13, miR-199a-5p	[251,281]
HOTAIR	12q13.13	Oncogenic	EMT-related genes, MMPs, miR-206/TBX3 axis, miR-138-5p/CHEK1	[117,118,120,121]
DLEU1	13q14.3	Oncogenic	miR-490-3p/CDK1 expression	[282,290]
EIBC	_	Oncogenic	Wnt/β-catenin	[97]
MALAT1	11q13.1	Oncogenic	YAP, Notch1 signaling pathway, miR-143-3p/CMPK, miR-503-5p/JAK2-STAT3, miR-200c, EMT via RBFOX2, KIF1B, β-catenin, DVL2, cyclin D1, Wnt/β-catenin signaling pathway	[182,207,208,209,210,211,212]
MNX1-AS1	_	Oncogenic	CDK4, cyclin D, BCL-2, BAX	[97]
MEG3	14q32.3	Tumor suppressor	PTEN, miR-205-5p, miR-219a-5p/EGFR axis, miR-421/PDGFRA axis, Notch1 pathway	[242,243,244,245,246]
SNHG15	7p13	Oncogenic	miR-18a, AKT/mTOR signalling pathway	[264]
XIST1	Xq13.2	Tumor suppressor	miR-150-5p	[264]
Linc-ROR	18q21.31	Oncogenic	EMT via Wnt/β-catenin signaling	[291]
NBAT1	6p22. 3	Tumor suppressor	ERK1/2, Akt pathways	[279]
UCA1	19p13.12	Oncogenic	miR-129/ABCB1 axis, SRPK1	[259]
lncBRM	_	Oncogenic	Sox4, miR-204	[97]
H19	11p15.5	Oncogenic	EMT via miR-370-3p/TGF-β pathway, IGF2	[165,167]
ZFAS1	20q13.13	Oncogenic	miR-548e, let-7a, E-cadherin, N-cadherin CXCR4, Vimentin, MMP-2, BCLXL, miR-150-5p, KLF2,	[263]
PCAT1	8q24	Oncogenic	miR-129-5p, cyclin D1/CDK4, NEK2/Wnt pathway, miR-124-3p/cyclin D1, CDK6, p53, BAX, cleaved caspase-3, metallopeptidases, vimentin, Wnt3a, β-catenin	[270]

**Table 5 cancers-13-06102-t005:** LncRNAs involved in proliferation, invasion, apoptosis, migration, metastasis, and drug resistance in vulvar cancers.

LncRNAs	Locus	Status	Target/Function	References
HOTAIR	12q13.13	Oncogenic	Unknown	[122]
MALAT1	11q13.1	Oncogenic	Unknown	[213,214]
MIR31HG	9p21.3	Oncogenic	p16INK4A	[292]
NEAT1	11q13.1	Oncogenic	Unknown	[144]
ROCK1	_	Oncogenic	Unknown	[293]
UCA1	19p13.12	Oncogenic	miR-103a/WEE1	[294]

**Table 6 cancers-13-06102-t006:** LncRNAs involved in proliferation, invasion, apoptosis, migration, metastasis, and drug resistance in GTN.

LncRNAs	Locus	Status	Target/Function	References
OGFRP1	22q13.2	Oncogenic	AKT/mTOR	[295]
LINC00261	20p11.21	Tumor suppressor	Unknown	[295]
MALAT1	11q13.1	Oncogenic	miR-218/Fbxw8	[215]
PCA3	9q21-22	Oncogenic	miR-106b	[295]
MEG3	14q32.3	Tumor suppressor	miR-211/PI3K/AKT and AMPK pathways, PI3K/AKT/NF-κB signaling pathway	[247,248]
MIR503HG	Xq26	Tumor suppressor	Unknown	[295]
H19	11p15.5	Oncogenic	PI3K/AKT/mTOR	[170]
LOXL1-AS1	_	Tumor suppressor	miR-515-5p/NF-κB signaling pathway	
SPRY4-IT1	_	Oncogenic	EMT process	[296]

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
