# Peer review of "The Role of Long Non-Coding RNAs (lncRNAs) in Female Oriented Cancers"

_cancers, 2021, doi:10.3390/cancers13236102_

Round 1

Reviewer 1 Report

The authors have modestly improved their manuscript by adding more content into the review. However, I can still easily find a large number of grammatical errors and broken poorly-written sentences (some listed below), and the authors should pay serious efforts to improve their writing and proofreading (or consider language-editing services). Right now the manuscript is hardly readable.

  1. The directions of the texts in the figures are not uniformly upright (e.g. many are flipped). This makes the figures very hard to read.
  2. Table 1 should be reformatted into several smaller tables, as it now spans 6 full pages.
  3. Line 72: ruling out? Do the authors intend to mean identify here?
  4. Line 194: lncRNAs “are” classified into…..
  5. Line 203: “Such as” should be “For example”
  6. Line 214: “lincRNA” typo
  7. Line 236: activate should be “activation of”
  8. Line 238: remove “While”
  9. Line 264: based on their requirement? What requirement?
  10. Line 289: another “such as” that should be “for example)
  11. Line 388: is expressed
  12. Line 596: induces angiogenesis by downregulating VEGF? Should it be upregulating?

,..

There are many more grammatical errors throughout the manuscript which should be corrected.

Reviewer 2 Report

I do not feel that my concerns have been addressed. I find the changes introduced by the authors rather superficial. I strongly belive the text requires additional modifications according to the lines I mentioned in my previous reviewer comments. 

Reviewer 3 Report

I congratulate the authors for incorporating the modifications. The manuscript now is in a better format and much closer to publication. However, I request to take note of the following minor points:

  1. Each section should end with brief summary and future directions (1-3 lines).
  2. The section on line 302- please mention explicitly the kind of therapy. For instance, which drug was given as chemo?
  3. The section on line 334- authors have added a wealth of information. I just have a concern, while describing the studies involving human patients, authors should quote the number of patients and the statistical inference (HR, P-value) observed.
  4. Reference 73- please specify the silenced genes.

Reviewer 4 Report

The addition of Table 1 greatly enhances the content of the review.  

Scheme 1, Figures 4-8- The placement of labels upside down or sideways is awkward.  The reader should not have to rotate their head to read. 

Round 2

Reviewer 1 Report

English is much better now. I am OK with acceptance.

Reviewer 2 Report

The authors have addressed my comments. 

This manuscript is a resubmission of an earlier submission. The following is a list of the peer review reports and author responses from that submission.

Round 1

Reviewer 1 Report

  1. I feel that this review is simply listing the different types of female cancers (epidemiology), followed by a rather repetitive listing of different literature studies of lncRNA with no insightful discussion. I think a useful review should be more than this.
  2. Within such a short review, more than 330 references are used. The authors should be more selective and concise when citing literature.
  3. The authors should proofread the entire manuscript again as there are many grammatical errors (e.g. faulty use of the plural s) and broken sentences here and there (some listed below).

Line 79-81: confusing sentence. “wither”?

Line 107-109: confusing sentence and possibly incorrect statement.

Line 133: “gynecological malignancies” is too broad. If you mean gynecological cancers, state gynecological cancers here.

Line 138: distinctive features of PFTC, maybe could briefly name a few?

Line 143-144: broken sentence. Please reword.

Line 192: can decreases? Can decrease

Line 221-223: A sentence that should be deleted.

Line 234: got a release?

Line 237-239: broken sentence.

Line 244: intrgenic?

Line 265-266: enticing and obliging are two confusing words. Should be replaced.

Line 268-269: most important lncRNAs? Selected as “most important” based on what? In figure 3, there are a load of lncRNAs.

Line 281: is overexpressed

Line 288: broken sentence

Line 310: broken sentence

And many others not yet listed here.

Reviewer 2 Report

Overall, this is a well written review on the roles of lncRNAs in female oriented cancers. The authors' overview of the current literature is rather comprehensive and the text is logically structured. Despite this, I strongly believe the text needs an additional "summary" section where the authors should carry out the synthesis of all the information they have provided, integrate these facts in the single picture and, possibly speculate on the unresolved questions and controversies in the field. I do not see a lot of point in publishing this review without such an analysis. 

The text should be checked for typos/English once again. At the moment it sometimes contains some unrelated parts - e.g. "circular intronic RNAs (ciRNAs) or both intron and exon fragments of the parent gene This section may be divided by subheadings. It should provide a concise and precise de-221 scription of the experimental results, their interpretation, as well as the experimental con-222 clusions that can be drawn. (elciRNAs). Such circRNAs are seen to be more stable as com-223 pared to linear" 

Reviewer 3 Report

In the present review, the authors comprehensively discuss the role of lncRNAs in female cancers and highlight the downstream mechanisms. I have several reservations. My comments are appended as below:

  1. The title should be reframed- ‘all in one’ looks vague.
  2. Please discuss the role of lncRNAs in association with age, menopause status.
  3. The authors should include a section on cross-talk between lncRNAs and hormone signaling prevalent among women.
  4. Please do not use vague terms throughout (for example, line 13, 42).
  5. Please elaborate on the nuclear effect of lncRNAs in cancer (epigenetic/ chromatin modifications/methylations- activating and inhibitory, etc).
  6. Status of cancers in females: please include median survival observed among pathological types.
  7. Line 204- please discuss the enhancing effects on miRNAs as there is very little literature available on this.
  8. Figure 2- back and green lines looks confusing.
  9. Table 1- please include if the study involves human subjects. Details such as no of patients, statistical inference (HR, P-value) should be added.
  10. Figure 3- if mechanisms are clearly reported, please include a table on it.
  11. Please include a section on therapy resistance and lncRNAs in cancer.
  12. Authors should discuss the presently available therapies on lncRNAs. This should include FDA-approved drugs or drugs under trial.
  13. Please include the future directions section.

Reviewer 4 Report

The review desperately needs assistance in writing (English grammar).  There are so many areas where editing is needed that a recommendation is to bring on an author familiar with the subject to assist in rewriting the manuscript.  

The section on status of cancers in female(s) appears to be focussed on incidence of the cancers, of which the connection to the main topic of long noncoding RNAs(LncRNAs) is unclear.  In the LncRNA section 3, the focus appears to focus on a few well known LncRNAs such as NEAT1, MALAT1, H19, HOTAIR, (M)EG3 and the rest of the LncRNAs appear to be left to Figure 3 with little discussion of any.  Hence the direction of the review (LncRNAs) is confusing and needs focus.